# Hybrid-control arm construction using historical trial data for an early-phase, randomized controlled trial in metastatic colorectal cancer

Chen Li [1✉], Ana Ferro [1], Shivani K. Mhatre[2], Danny Lu[3], Marcus Lawrance[1], Xiao Li[2], Shi Li[2], Simon Allen[2], Jayesh Desai[4], Marwan Fakih[5], Michael Cecchini[6], Katrina S. Pedersen[7], Tae You Kim[8], Irmarie Reyes-Rivera[9], Neil H. Segal[10] & Christelle Lenain[9✉]

## Abstract

**Background** Treatment for metastatic colorectal cancer patients beyond the second line remains challenging, highlighting the need for early phase trials of combination therapies for patients who had disease progression during or following two prior lines of therapy. Leveraging hybrid control design in these trials may preserve the benefits of randomization while strengthening evidence by integrating historical trial data. Few examples have been established to assess the applicability of such design in supporting early phase metastatic colorectal cancer trials.

**Methods** MORPHEUS-CRC is an umbrella, multicenter, open-label, phase Ib/II, randomized, controlled trial (NCT03555149), with active experimental arms ongoing. Patients enrolled were assigned to a control arm (regorafenib, 15 patients randomized and 13 analysed) or multiple experimental arms for immunotherapy-based treatment combinations. One experimental arm (atezolizumab + isatuximab, 15 patients randomized and analysed) was completed and included in the hybrid-control study, where the hybrid-control arm was constructed by integrating data from the IMblaze370 phase 3 trial (NCT02788279). To estimate treatment efficacy, Cox and logistic regression models were used in a frequentist framework with standardized mortality ratio weighting or in a Bayesian framework with commensurate priors. The primary endpoint is objective response rate, while disease control rate, progression-free survival, and overall survival were the outcomes assessed in the hybrid-control study.

**Results** The experimental arm showed no efficacy signal, yet a well-tolerated safety profile in the MORPHEUS-CRC trial. Treatment effects estimated in hybrid control design were comparable to those in the MORPHEUS-CRC trial using either frequentist or Bayesian models.

**Conclusions** Hybrid control provides comparable treatment-effect estimates with generally improved precision, and thus can be of value to inform early-phase clinical development in metastatic colorectal cancer.

## Plain language summary

Treatment of patients with metastatic colorectal cancer – meaning that it has spread to other parts of the body – is difficult, and new therapies are needed for patients when standard therapies stop working. We compare a combination of drugs with a standard treatment for patients with metastatic colorectal cancer in a clinical trial, in which patients are randomly allocated to either the combination or the control (standard) treatment. We find that while the combination is safe, it isn't effective. We also show, however, that we can combine data from our control group and the control group of a previous trial to more precisely estimate treatment effects. Statistical approaches such as this to combine data from trials may mean that fewer patients have to be recruited to control groups in future trials, to improve access to potentially effective new treatments.

[1] Roche Products Limited, Welwyn Garden City, UK. [2] Genentech, Inc., South San Francisco, CA, US. [3] Hoffmann-La Roche Limited, Mississauga, ON, Canada. [4] Peter MacCallum Cancer Centre, Melbourne, VIC, Australia. [5] City of Hope Comprehensive Cancer Center, Duarte, CA, USA. [6] Yale Cancer Center, New Haven, CT, USA. [7] Washington University School of Medicine, St. Louis, MO, USA. [8] Seoul National University College of Medicine, Seoul, South Korea. [9] F. Hoffmann-La Roche Ltd, Basel, Switzerland. [10] Memorial Sloan Kettering Cancer Center, New York City, NY, USA. ✉email: chen.janie.li@outlook.com; christelle.lenain@roche.com

Randomized controlled trials are regarded as the gold standard for evaluating effectiveness of treatments, yet regulatory agencies are becoming more receptive to supplementing or replacing a control arm with historical data from previously completed trials, especially in rare and pediatric diseases or for life-threatening cancer indications with few treatment options[1,2]. In such scenarios, randomizing patients to control arms may be less acceptable due to ethical or feasibility considerations, leading to a higher proportion of patients dropping out when randomized to control arms or less likely to consent if there are higher odds of being randomized to control arms[3]. Moreover, even in trials of more-prevalent diseases or with specific eligibility criteria, challenges may be found during patient recruitment—for instance, in late-stage cancer trials with requirements for specific biomarker status[4]. Hybrid-control design using relevant individual patient data from historical clinical trials is being explored as a way to achieve more patient-centric, cost-effective, and accelerated clinical development, since fewer patients are needed for standard-of-care or placebo-control arms[5,6]. How to determine the amount of borrowing for the control arm is based on comparability between historical- and concurrent-control arms, a key question for implementing a hybrid-control design[7], and few examples have been established to assess the applicability of such design in supporting early-trial development.

Treatment for metastatic colorectal cancer (mCRC) patients beyond the second line remains challenging, despite the success of single-agent checkpoint inhibition in the patient population with microsatellite instability-high status. mCRC patients are mostly microsatellite stable, and thus do not respond to the single-agent checkpoint inhibition, highlighting the need for combination therapies. The MORPHEUS platform consists of multiple randomized umbrella phase 1b/2 trials designed to identify early efficacy signals in small cohorts and accelerate development of treatment combinations across a wide scope of cancer indications[8]. In the MORPHEUS mCRC trial, patients with microsatellite stable tumors who had been refractory to the first and second line of therapies were randomized to either experimental arms or a control arm with regorafenib, a standard-of-care therapy in this disease setting. The relatively small sample size inherent to early-phase trials can limit their potential to detect a treatment effect. Here we report the primary results of the experimental arm (atezolizumab + isatuximab) and the control arm (regorafenib) and investigate the hybrid control trial design with data integrated from historical control arm data of the IMblaze370 trial. The combination of atezolizumab plus isatuximab lacks efficacy, while the safety profile of the experimental arm is consistent with that of the control arm. The use of hybrid control design improves precision while maintains accuracy of estimates from a randomized trial.

## Methods

**Study design**. This study established a hybrid-control arm for the MORPHEUS-CRC trial using historical control data from the IMblaze370 trial. MORPHEUS-CRC (ClinicalTrials.gov Identifier: NCT03555149) is an ongoing, phase 1b/2, open-label, multicenter, randomized study designed to identify early signals of safety and efficacy of immunotherapy-based treatment combinations in patients with refractory microsatellite-stable mCRC[8,9]. Patients in the MORPHEUS-CRC trial were randomly assigned to different treatment arms with a permuted-block randomization method; study sites obtained patients' identification numbers and treatment assignments from an interactive voice or web-based response system (IxRS). The control arm (regorafenib) and the experimental arm (atezolizumab + isatuximab) were included in

this study and were enrolled between September 2018 and August 2019 (Supplementary Fig. 1). IMblaze370 (ClinicalTrials.gov Identifier: NCT02788279) is a completed, phase 3, multicenter, open-label, randomized trial study that enrolled patients with mCRC who had disease progression with at least 2 previous systemic chemotherapy regimens between July 2016 and January 2017[10]. Patients in the IMblaze370 control arm who met the MORPHEUS-CRC eligibility criteria were selected to build an external-control arm (Supplementary Note 2). A detailed comparison of eligibility criteria in the IMblaze370 and MORPHEUS-CRC trials is in Supplementary Data File 1. The external-control arm was incorporated into the MORPHEUS-CRC concurrent-control arm to construct a hybrid-control arm using a frequentist model with propensity score (PS) weighting or a Bayesian dynamic borrowing method. An overview of the study design is shown in Fig. 1. MORPHEUS-CRC trial was reviewed by the institutional review board at each site (Supplementary Note 4), as well as the IMblaze370 trial[10]. All participants provided informed written consent. The present hybrid-control study was not pre-specified in the MORPHEUS-CRC trial protocol. The study followed the Consolidated Standards of Reporting Trials (CONSORT) reporting guideline (Supplementary Fig. 3).

**Outcome assessment**. Investigator-assessed objective response rate (ORR) was the primary endpoint for MORPHEUS-CRC. For hybrid control analyses, the key secondary endpoints of disease control rate (DCR), investigator-assessed progression-free survival (PFS), and overall survival (OS) were the outcomes evaluated. ORR was not evaluated in the hybrid control analyses, because no response was observed in either of the arms. DCR was defined as the proportion of patients with complete or partial response at any time during the trial or stable disease for at least 12 weeks in the MORPHEUS-CRC. Similar definition of DCR was applied in the IMblaze370 trial, but with stable disease for at least 16 weeks. As the time interval for response assessment was every 6 weeks in the MORPHEUS-CRC and every 8 weeks in the IMblaze370 trial, the DCR in the IMblaze370 trial at 12 weeks was inferred using tumor overall response assessment at 8 and 16 weeks. Specifically, if a patient showed progressive disease at 8 weeks in the IMblaze370 trial, then the response for that patient at 12 weeks was defined as progressive disease; if a patient showed stable disease at both 8 and 16 weeks, then the response for that patient at 12 weeks was defined as stable disease; otherwise, a patient's response was set as unknown. Disease progression was determined by clinical investigators according to the Response Evaluation Criteria in Solid Tumors (RECIST) v1.1[11]. PFS was defined as the time from trial randomization to the occurrence of disease progression or death (whichever occurs first) or end of trial follow-up. OS was defined as the time from trial randomization to the occurrence of death or end-of-trial follow-up. PFS and OS time in the external-control arm were truncated to match with the maximum PFS and OS time of the MORPHEUS-CRC trial, respectively. Safety was also reported for the MORPHEUS-CRC using the National Cancer Institute's The Common Terminology Criteria for Adverse Events, version 4.0.

**Propensity score estimation**. Potential imbalance of predefined baseline characteristic and prognostic factors (hereafter referred to as baseline covariates) between the MORPHEUS experimental arm and the external-control arm were adjusted using PS with standardized mortality ratio weighting (SMRW) method. The SMRW method was preferred to the inverse probability of treatment weighting (IPTW) method in this scenario, due to considerations given in the Supplementary Note 1. PS was estimated using a multivariate logistic regression model adjusted for

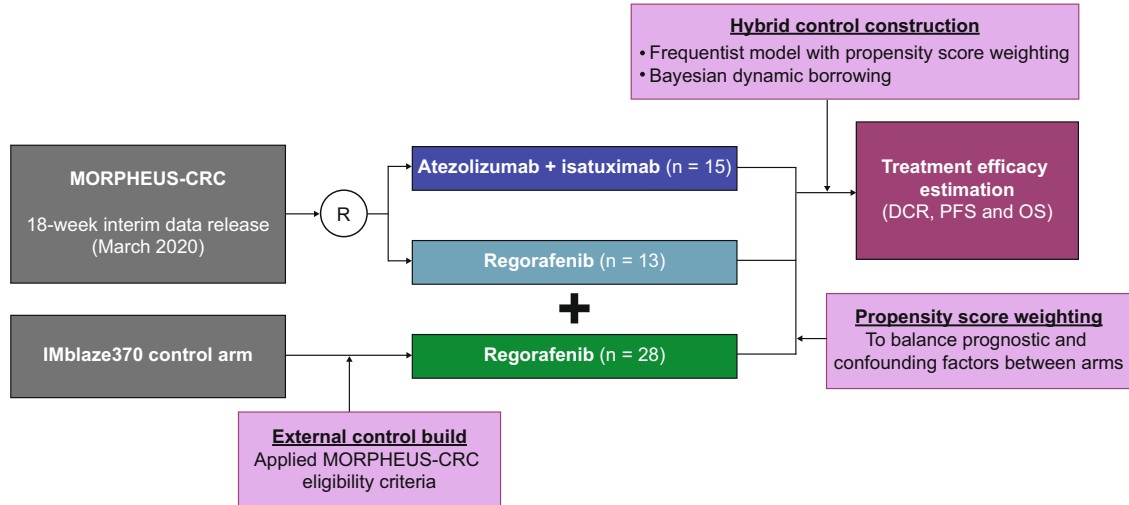

**Fig. 1 Study overview.** Patients from the IMblaze370 trial control arm (regorafenib) who received regorafenib as the third-line treatment and met the MORPHEUS-CRC trial eligibility criteria were selected for the external-control cohort and incorporated in the MORPHEUS concurrent control arm to construct a hybrid-control cohort. DCR disease control rate, EC external control, HC hybrid control, I/E inclusion/exclusion, mCRC metastatic colorectal cancer, OS overall survival, PFS progression-free survival, PS propensity score, R randomization.

predefined covariates. This model assumed a linear relationship between each baseline covariate and the log-odds of the group assignment (being in the MORPHEUS-CRC experimental arm vs external-control arm). PS was calculated for each patient, representing a patient's probability of being in the MORPHEUS-CRC experimental arm, conditioning on all baseline covariates.

The baseline covariates selected included age, sex, presence of liver metastasis, time from metastatic diagnosis to baseline (> vs ≤18 months), and Eastern Cooperative Oncology Group (ECOG) performance score (0 vs 1). Covariate selection was based on data availability, model convergence, and potential clinical importance with respect to their prognostic impact in the metastatic refractory setting[12]. Balance was assessed with standardized mean difference (SMD), where covariates with SMD < 0.25 were deemed as sufficiently balanced[13,14].

### Hybrid-control modeling

*Bayesian borrowing.* Combining randomized- and historical-control arms in a Bayesian framework allows a dynamic proportion of the historical-control arm to be used in the hybrid-control arm in a data-driven manner. The proportion was determined by commensurability between the external-control and the concurrent-control arm. This was derived firstly based on one's subjective determination via the prior setting on a value for the variance of difference between mean treatment-effect sizes of the two control arms, then updated with data likelihood, to produce a posterior belief for the proportion of borrowing[7]. A prior setting for the variance incorporates one's initial guidance for the degree of borrowing; by increasing values of the prior, one places more emphasis on the randomized control and less on the historical control, i.e., discouraging the borrowing, and vice versa.

For the DCR analyses, given there were only two levels observed, stable disease (SD) and progressive disease (PD), we assumed DCR for a patient $i$ ($y_i$) to follow a Bernoulli distribution $y_i \sim Bernoulli(p_i)$, with $p_i$ referring to the probability of SD for the patient $i$. We defined $\gamma0$, $\gamma1$, and $\gamma2$ to be the logit function of $p_i$ in the MORPHEUS-CRC experimental arm, the concurrent-control arm, and the external-control arm, respectively; $\gamma1$ follows a normal distribution with a mean of $\gamma2$ and a variance of $1/\tau$, where $\tau$ follows a gamma(1,1) distribution; $\gamma0$ and $\gamma1$ are both non-informative vague priors with the Gaussian normal distribution.

For the PFS and OS analyses, we assumed survival time for a patient $i$ ($t_i$) to follow a Weibull distribution $t_i \sim Weibull(r, \mu_i)$, with a shape of $r \sim \exp(10)$, and a scale parameter of $\mu_i$. By setting the natural logarithm of hazards to be $\beta0$, $\beta1$, and $\beta2$ in the MORPHEUS-CRC experimental arm, the concurrent-control arm, and the external-control arm, respectively, we derived $\mu_i = e^{\beta0}$, $e^{\beta1}$, $e^{\beta2}$ in each corresponding arm; $\beta1$ follows a normal distribution with a mean of $\beta2$ and a variance of $1/\tau$, where $\tau$ parametrizes commensurability and determines the degree of borrowing, which follows a half-Cauchy (0, 25) distribution; $\beta0$ and $\beta1$ are set to be noninformative, following standard normal distributions[15].

When evidence for commensurability is weak, $\tau$ is forced toward zero, increasing the prior variance of $\beta1$ by $1/\tau$, thereby discouraging borrowing from external data[5]. To assess the potential impact of the prior choices of $\tau$ on the results, we performed sensitivity analyses with different $\tau$ distributions. For the PFS and OS, results were consistent across different prior distributions; for the DCR, results changed slightly with priors ($1/\tau$) of larger variances, partially due to the very small sample size and thus very large standard deviations of the result estimates (Supplementary Table 1). Moreover, we assessed the amount of prior-data conflict by visualising prior and posterior distributions of $\beta$ coefficients for all three outcomes (Supplementary Fig. 2).

The implementation was written in JAGS[16] using Markov chain Monte Carlo with 3 parallel chains, each run for a 1000-iteration burn-in period followed by a 20,000-iteration production run.

*Frequentist with SMRW.* Logistic regression was implemented for the binary endpoint of DCR; Cox proportional hazards models were implemented for the time-to-event endpoints, PFS and OS; each model contained only one exposure variable, the group of treatment (experimental vs control treatment), and was weighted with SMRW to balance baseline covariates through the PS method. Robust variance estimator was applied for weighted models to account of within-subject correlations in the weighted pseudo-population, because a lack of independence between subjects can cause a naive model-based variance estimator more likely to be biased, and such robust method has been shown to be an option for unbiased variance estimation in this setting[17].

**Statistical analysis.** Baseline demographic and clinical characteristics were summarized in the external-control (regorafenib)

arm derived from the IMblaze370 trial, and the MORPHEUS-CRC concurrent-control (regorafenib) and experimental (atezolizumab + isatuximab) arms, separately. Experimental treatment efficacies were estimated by comparing the MORPHEUS-CRC experimental arm to the concurrent-control or the hybrid-control arm in a frequentist or a Bayesian framework. Survival time (PFS and OS) was determined using the Kaplan–Meier estimator with SMRW, with median point estimates and corresponding 95% CIs summarized for each arm along with the Kaplan–Meier curves. All analyses were conducted using RStudio version 1.3.0 and R version 3.6.3.

**Reporting summary**. Further information on research design is available in the Nature Research Reporting Summary linked to this article.

## Results

**Primary MORPHEUS-CRC results**. No responses occurred in either the experimental arm–atezolizumab + isatuximab (n = 15) or the control arm–regorafenib (n = 13) in the MORPHEUS-CRC trial. DCR was 13.3% in the experimental arm and 15.4% in the control arm. Median PFS was 1.4 months (95% CI: 1.4–1.8) in the experimental arm and 2.8 months (95% CI: 1.6–3.1) in the control arm; median OS was 5.1 months (95% CI: 3.1–7.8) and 10.2 months (95% CI: 4.8, not reached estimable) in the experimental and the control arms, respectively (Table 1). A summary of the safety data can be found in Supplementary Tables 2–3. Overall, the atezolizumab + isatuximab combination was well tolerated, with a manageable safety profile. The routine laboratory measurements at and post baseline were comparable and, for both arms, mostly normal or of the minimum grade of severity according to The Common Terminology Criteria for Adverse Events. No new safety signals were observed.

**Study population**. The external-control cohort was derived from the IMblaze370 control arm who received regorafenib (n = 90). Cohort attrition using the MORPHEUS-CRC eligibility criteria (Supplementary Note 2) as filtering steps yielded 28 patients who constituted the external-control group (Fig. 2). There was no major imbalance in the baseline demographic or clinical characteristics, including age, sex, race, time from metastatic diagnosis to baseline, ECOG performance score, *Rat Sarcoma (RAS) proto-oncogene* mutational status, and presence of liver metastases (Table 2). Regarding the region, the external-control patients were mainly from Europe, whereas the MORPHEUS-CRC patients were from North America, Asia, or Australia.

**Balance of confounding variables**. To account for potential differences between the predefined baseline covariates in the two trials, we leveraged the PS approach, which is commonly used for causal inference on treatment effects in observational studies[18] or single-arm trials supplemented with external-control design[19–21]. Using this method, the external-control cohort was further balanced on a selected subset of baseline variables, including age, sex, time from metastatic diagnosis to baseline, ECOG performance score, and presence of liver metastases, with each variable demonstrating an acceptable balance with SMD < 0.25 (Supplementary Fig. 4).

*Efficacy assessment*. Three endpoints were used for treatment efficacy assessment: DCR, PFS, and OS. Five patients with unknown response assessment were excluded from the external dataset in the DCR analyses (Supplementary Table 4).

**Table 1 MORPHEUS-CRC Treatment Efficacy.**

| | Atezolizumab + isatuximab (n = 15) | Regorafenib (n = 13) |
|---|---|---|
| Confirmed ORR, No. (%) | 0 (0.0) | 0 (0.0) |
| % [95% CI]ª | [0, 21.8] | [0, 24.7] |
| SD, No. (%) | 3 (20.0) | 8 (61.5) |
| % [95% CI]ᵇ | [4.3, 48.1] | [31.6, 86.1] |
| PD, No. (%) | 10 (66.7) | 3 (23.1) |
| % [95% CI]ᵇ,ᶜ | [38.4, 88.2] | [5.0, 53.8] |
| DCR, No. (%) | 2 (13.3) | 2 (15.4) |
| % [95% CI]ᵈ | [1.7, 40.0] | [1.9, 45.5] |
| PFS, median survival (months, 95% CI)ª | 1.4 (1.4–1.8) | 2.8 (1.6–3.1) |
| OS, median survival (months, 95% CI)ª | 5.1 (3.1–7.8) | 10.2 (4.8-NE) |

Clinical cutoff: 3 March 2020. *No.* number of patients, *DCR* disease control rate, *NE* not estimable, *ORR* objective response rate, *OS* overall survival, *PD* progressive disease, *PFS* progression-free survival, *RECIST* Response Evaluation Criteria in Solid Tumors, *SD* stable disease, *CI* confidence interval.
ªPer INV RECIST 1.1.
ᵇPatients were classified as achieving stable disease or progressive disease if assessment was at least 6 weeks from randomization.
ᶜOne patient treated with atezolizumab + isatuximab beyond progression had prolonged disease stabilization.
ᵈCriteria for disease control is either response and/or stable disease for at least 12 weeks.

In the original MORPHEUS-CRC trial, these three endpoints favored the regorafenib arm, although the differences were not statistically significant. Using the hybrid control design in a frequentist framework showed comparable estimates, with substantial improvements in precision for all three endpoints (Fig. 3).

Hybrid-control design with a Bayesian framework also showed comparable estimates for all three endpoints. Precisions were improved for the PFS and the OS, but not the DCR, indicating a higher degree of commensurability between the 2 control arms for the PFS and the OS (HR [95% CI] = 1.04 [0.67–1.63], $\tau_{mean} = 3.52 \times 10^5$; HR [95% CI] = 1.00 [0.66–1.38], $\tau_{mean} = 7.48 \times 10^5$ for the PFS and the OS, respectively, comparing hybrid-control with concurrent-control arms, Supplementary Fig. 5), leading to an effective borrowing of power. This was not the case for the DCR, as the amount of borrowing was restricted due to a larger amount of dissimilarity between the two control arms (OR [95% CI] = 0.83 [0.20–1.97], $\tau_{mean} = 13.15$, comparing the hybrid control with the concurrent-control arms, Supplementary Table 1).

## Discussion

In this study, we constructed a hybrid-control arm using the historical trial (IMblaze370) data to supplement a concurrent randomized controlled trial (MORPHEUS-CRC) on treatment efficacy estimation based on three endpoints (DCR, PFS, and OS). Comparison using the hybrid-control or the concurrent-control arm led to similar experimental treatment-effect estimates across the endpoints assessed, with the hybrid-control design generally achieving greater precisions.

The degree of borrowing from the external-control cohort depends on the similarity between the external-control and the concurrent-control arm, and assessment of the similarity and adjusting accordingly differentiates hybrid-control methods[22]. For example, novel methods that use power priors[23] or meta-analytic predictive priors[24] with PS-based selection of patients can assess various degrees of between-trial heterogeneity, and adaptively adjusting the amount of borrowing of external information. Here we leveraged two methods: the first was the static borrowing method under the frequentist paradigm, and the second was the dynamic borrowing method within the Bayesian framework. For

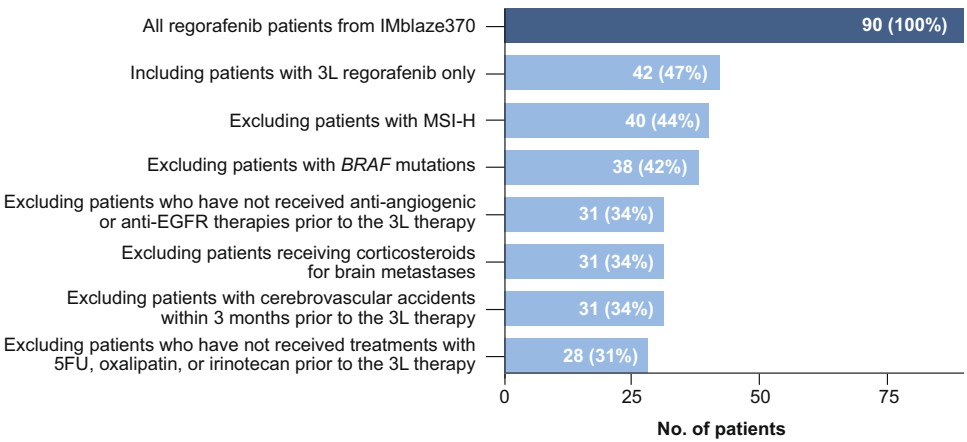

**Fig. 2 IMblaze370 control cohort attrition.** The external-control cohort was built with the MORPHEUS trial eligibility criteria applied in the IMblaze370 control cohort. 3 L third-line, 5FU fluorouracil, BRAF B-Raf proto-oncogene, serine/threonine kinase, EGFR epidermal growth factor receptor, MSI-H microsatellite instability–high.

**Table 2 Comparison of the Baseline Demographic and Disease Characteristics Between the External-Control Arm Derived From the IMblaze370 and the MORPHEUS-CRC Control and Experimental Arms.**

| | IMblaze370 | MORPHEUS-CRC | | P value (EC vs Atezo + Isa) | SMD (EC vs Atezo + Isa) |
|---|---|---|---|---|---|
| | Regorafenib (EC) | Atezo + Isa | Regorafenib | | |
| Total Sample size | 28 | 15 | 13 | | |
| Age at baseline, mean (SD) | 57.0 (9.6) | 52.2 (12.0) | 59.5 (10.5) | 0.178 | 0.445 |
| Sex, No. (%) | | | | 0.859 | 0.058 |
| Female | 12 (42.9) | 6 (40.0) | 7 (53.8) | | |
| Male | 16 (57.1) | 9 (60.0) | 6 (46.2) | | |
| Race, No. (%) | | | | 0.362 | 0.306 |
| White | 21 (84.0) | 10 (71.4) | 8 (66.7) | | |
| Non-White | 4 (16.0) | 4 (28.6) | 4 (33.3) | | |
| Unknown | 3 | 1 | 1 | | |
| Region, No. (%) | | | | 0.002 | 1.498 |
| North America | 6 (21.4) | 11 (73.3) | 6 (46.2) | | |
| Europe | 17 (60.7) | 1 (6.7) | 2 (15.4) | | |
| Asia-Pacific | 5 (17.9) | 3 (20.0) | 5 (38.5) | | |
| Time from metastatic diagnosis to baseline, No. (%) | | | | 0.161 | 0.464 |
| <18 months | 7 (25.0) | 7 (46.7) | 4 (30.8) | | |
| ≥18 months | 21 (75.0) | 8 (53.3) | 8 (61.5) | | |
| Unknown | 0 | 1 | 0 | | |
| ECOG, No. (%) | | | | 0.417 | 0.27 |
| 0 | 13 (46.4) | 5 (33.3) | 6 (46.2) | | |
| 1 | 15 (53.6) | 10 (66.7) | 7 (53.8) | | |
| RAS, No. (%) | | | | 0.924 | 0.032 |
| Wild type | 10 (38.5) | 6 (40.0) | 8 (61.5) | | |
| Mutant | 16 (61.5) | 9 (60.0) | 5 (38.5) | | |
| Unknown | 2 | 0 | 0 | | |
| Liver metastases, No. (%) | | | | 0.786 | 0.088 |
| No | 10 (35.7) | 6 (40.0) | 4 (30.8) | | |
| Yes | 18 (64.3) | 9 (60.0) | 9 (69.2) | | |

Statistical differences between the external-control and the MORPHEUS experimental arm (atezolizumab + isatuximab) were assessed using (1) P values calculated via the 2-tailed χ2 (or Fisher exact) test for all categorical variables or the Wilcoxon rank-sum test for the age variable, and (2) standardized mean difference.
*Atezo* atezolizumab, *EC* external control, *ECOG* Eastern Cooperative Oncology Group, *Isa* isatuximab, *RAS* Rat Sarcoma proto-oncogene.

the first method, the external-control cohort was balanced for baseline covariates using SMRW and a frequentist model was subsequently applied for treatment-effect estimates. The resultant weighted pseudo-cohort can be regarded as a statistically balanced external-control arm that is able to merge with the concurrent-control arm to form the hybrid-control arm. For the second method, a commensurate prior was used to assess how comparable the external-control arm was to the randomized-control arm. The posterior distribution of the commensurate prior is determined via a data-driven process, which further informs the degree of borrowing. Each method has its pros and cons, as discussed below; we also compared each method in a simulation framework with known parameters to reveal which method has more desirable statistical properties (Supplementary Note 3).

In this study we observed that the frequentist approach with SMRW was generally more effective for the precision improvement of treatment-effect estimates; however, it is more likely to be biased owing to residual imbalance of a priori selection of confounding factors or unmeasured confounding. We adjusted for five clinically relevant covariates, which were prioritized based on their potential influence on prognosis in the metastatic CRC refractory setting. Other factors with potential prognostic impact on mCRC were not included due to data incompleteness, model nonconvergence, or discrepancies of data collection or trial design between the historical trial and the MORPHEUS-CRC trial. Among the factors excluded were *RAS* mutation status and tumor sidedness. Although these factors appear to be prognostic at the time of diagnosis and initial treatment of mCRC[25], their

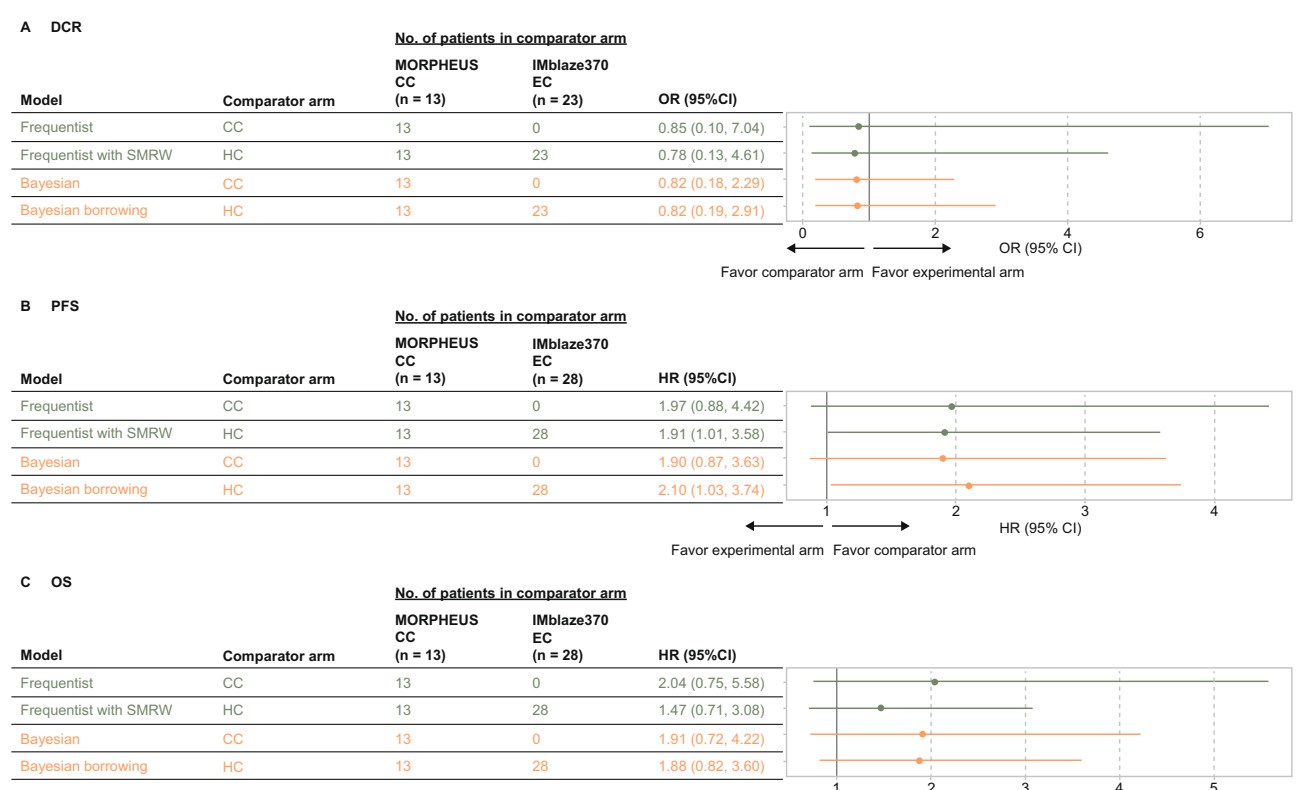

**Fig. 3 Treatment-effect estimates derived from experimental arm vs concurrent control arm or hybrid control arm.** Treatment efficacy was assessed via (**A**) DCR, (**B**) PFS, and (**C**) OS endpoints comparing the MORPHEUS experimental arm (atezolizumab + isatuximab) to the concurrent-control arm or the hybrid-control arm (regorafenib), separately. CC concurrent control, DCR disease control rate, EC external control, HC hybrid control, HR hazard ratio, OR odds ratio, OS overall survival, PFS progression-free survival, SMRW Standardized mortality ratio weighting.

prognostic effect in the refractory setting is unclear. Nonetheless, the covariates taken into consideration for confounding adjustment may be insufficient to achieve a complete balance between the external- and the concurrent-control arms, leading to a potential risk for bias that is inherited by the subsequent frequentist model-based experimental treatment-effect estimation.

The Bayesian dynamic borrowing, in contrast, can detect potential dissimilarities and adapt the amount of borrowing based on estimation of the commensurability parameter. Therefore, if the externally derived and the concurrent control have similar endpoint distributions, then a larger amount of the external control data is incorporated, leading to a substantially narrowed credible set, and vice versa. However, when the trial sample size is small, model inferences are heavily sensitive to prior settings[7]. In our analyses, the prior distributions for the treatment effects (i.e., beta coefficients) were set as standard normal distributions for all three endpoints. Due to these relatively informative priors, the credible sets from the Bayesian models were narrower than the confidence intervals from frequentist models in analyses with the concurrent control. Moreover, we applied a weakly informative prior distribution on the commensurability parameter, which provided an advantage of borrowing at a conservative level to reduce inflation of type 1 error rate, at a cost of potentially over-attenuating the influence of historical control data on the treatment-effect estimation[26].

This study has several limitations. First, as with all early-phase trials, there is a limitation owing to a small sample size. Therefore, for causal inferences, PS models cannot handle a large number of covariates, leading to potentially inadequate adjustment of confounders. In consideration of this, we applied a Bayesian method independent of PS adjustment, which counteracts the limitations

suffered by the frequentist models with PS weighting. In turn, we also used the frequentist model to benchmark the prior settings for the Bayesian models. Second, despite similar trial populations and randomization of the historical and current trials, the historical trial's recruitment period was initiated ~2 years before the MORPHEUS-CRC. These noncontemporary cohorts potentially create heterogeneity between the external-control arm and the concurrent trial. For example, during the two-year period since the IMblaze370 trial initiation, optimization of regorafenib dosing[27] and improved toxicity management were able to be incorporated in the MORPHEUS-CRC trial, which may have translated into better outcomes in control patients in the MORPHEUS-CRC compared with those in the IMblaze370. This potential source of heterogeneity may not be fully addressed, although by design, any unknown differences between the two trials have been embedded in the Bayesian commensurability prior setting. Third, definitions of the clinical outcomes vary in terms of response assessment frequencies (6 vs 8 weeks), which has led to missing DCR values in the external-control cohort, whereas, for the PFS and the OS, such differences in the frequencies of outcome assessment exerted minimal effects because cumulative survival probability distributions between the external-control and the concurrent-control cohorts were comparable, especially after balancing the baseline covariates.

In conclusion, this proof-of-concept study shows that a hybrid-control design using historical trial data from the IMblaze370 recapitulates the results obtained with the randomized-control arm of the MORPHEUS-CRC trial, with generally greater precisions. This result suggests that hybrid-control design may be used in early phase trials to support more confident decision-making

to inform later-phase development. This study also demonstrates the feasibility of supplementing a concurrent-control arm with a completed clinical trial–derived external-control arm. Overall, the hybrid-control design has the potential to increase overall trial attractiveness by favoring randomization to potentially transformative experimental arms. This approach may inform and accelerate early-phase clinical development, although it is not a shortcut to an approval of clinical practice during the process of drug development. Further research is warranted to expand this work in the refractory mCRC setting as well as other cancer indications.

## Data availability

Qualified researchers engaged in rigorous, independent scientific research may request access to individual patient-level data upon request through https://www.roche.com/innovation/process/clinical-trials/data-sharing/request. Data from individual patients can be requested 18 months after relevant clinical studies being approved by the regulatory authorities or will not be developed further. Requests outside this scope will be considered on a case-by-case basis through enquiries via the https://vivli.org site. Further details on data sharing and how to request access to related clinical study documents are available at https://www.roche.com/innovation/process/clinical-trials/data-sharing/. Information on the clinical trials can be found on clinicaltrials.gov. The IMblaze370 trial is published[10].

## Code availability

Custom code for Bayesian dynamic borrowing is accessible at https://doi.org/10.5281/zenodo.6514629[28]. Code for other analyses, including external cohort attrition and propensity score analyses, is available upon request from the author via https://github.roche.com/PHC/PHC-619/.

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

## Acknowledgements

The authors would like to acknowledge the patients and their families, investigators and clinical study sites for the MORPHEUS-CRC trial team; the IMblaze370 trial team for data access and Louise Roberts for providing scientific review; Bena Lim, PhD from MediTech Media, for editorial support for this manuscript, provided by F. Hoffmann-La Roche Ltd. This study was sponsored by F. Hoffmann–La Roche Ltd. Sanofi S.A. provided drug supply under an agreement between F. Hoffmann–La Roche Ltd and Sanofi S.A.

## Author contributions

Concept and design: C.Li, A.F., S.K.M., M.L., S.L., S.A., I.R.R. and C.Lenain. Acquisition, analysis or interpretation of data: C.Li, A.F., S.K.M., M.L., C.Lenain. Drafting of the manuscript: C.Li Critical revision of the manuscript for important intellectual content: C.Li, A.F., S.K.M., M.L., S.A., J.D., M.F., M.C., K.S.P., T.Y.K., I.R.R., N.H.S. and C.Lenain. Statistical analysis: C.Li, D.L., X.L., S.L. Obtained funding: I.R.R., C.Lenain. Supervision: I.R.R., C.Lenain, N.H.S.

## Competing interests

C.Li, A.F., S.K.M., D.L., L.M., S.L., S.A., I.R.R. and C.Lenain are employees of Hoffmann-La Roche Ltd., the company that funded the study. M.F. has received honoraria from Amgen, is an advisor/consultant for Amgen, Array BioPharma, Bayer, GlaxoSmithKline, HalioDx, Incyte, Mirati, Pfizer, Seattle Genetics, Taiho and Zhuhai Yufan Biotech, is a speaker for Amgen and Guardant360 and received institution research funding Amgen, AstraZeneca and Novartis. M.C. has received honoraria from Eisai Co., Agios Pharmaceuticals and AstraZeneca and travel/accommodation/expenses from Genentech/Roche and is receiving research support as part of the Yale Cancer Center Calabresi Immuno-oncology Training Program supported by the National Cancer Institute of the National Institute of Health under Award Number K12 CA215110. All other authors have no competing interests to declare.
