## [Peer Review File · Communications Medicine]

Reviewers' comments:

Reviewer #1 (Remarks to the Author):

The topic of this paper by Li et al is extremely relevant in the frame of the challenges of clinical research in the era of precision medicine.

Indeed the availability of molecular tests able to identify small subgroups of patients with distinct molecular characteristics and hopefully targeted options opens the way to the fragmentation of the disease and the need to re-think about the traditional paradigm of drugs' development.

To this regard, I fully embrace the message of this paper, but I also think that from a clinical perspective a few remarks should be made and included in the manuscript.

First of all, as previously mentioned, I do believe that these hybrid-designs might be especially relevant when small subgroups are identified and treated according to the presence of actionable alterations. In this case, having multiple control arms is probably uselessly expensive. However, if we treat patients with the molecular alteration A with the anti-A targeted agent, then we should compare their outcome with the one of patients with the same molecular alteration A treated with a standard option in an historical cohort, especially if this alteration A has a prognostic and not only predictive impact. How should this point be tackled by the proposed approach?

I would also like to see more comments about the choice of factors to be included in the propensity score evaluation. For example in the specific case study presented in the manuscript, among variables included in the propensity score adjustment some of the factors with known impact on patients' prognosis are missing including the number of organs involved, the timing of metastases onset (synchronous vs metachronous), the number of lines of therapy previously received, the presence of peritoneal metastases. How should these variables be chosen and implemented in the propensity score?

Finally, it should be made very clear that efficacy results coming from early phase trials are preliminary by definition and the hybrid design approach may help deriving more clinically relevant information to plan the subsequent steps of drugs' development especially when the feasibility of large scale trials is hampered by the rarity of some specific conditions rather than a way to "jump" the steps of drugs development towards clinical practice.

Prof. Chiara Cremolini, MD PhD

Reviewer #2 (Remarks to the Author):

The authors report results from a randomized trial and use data from one arm of a completed trial to augment the concurrent control arm. As a first step, they select from the historical control arm those patients that would have fulfilled the current eligibility criteria. Then they use a frequentist and a Bayesian method for constructing the so-called hybrid control arm. In the frequentist approach, they use a SMRW propensity approach to incorporate the external data. In the Bayesian approach, the authors use the commensurate prior approach to borrow from the historical data.

The authors address a timely topic. They do not develop new methodology but they apply two existing approaches to a data set and compare the results. In a way, it seems that the authors compare fundamentally different approaches: in the frequentist approach with PS weighting, they take care that the current and the historical arms are similar concerning the patients in regard to the confounders they used for PS calculation. In the Bayesian approach, they borrow irrespective of patient characteristics but adapting to the degree of similarity of the endpoints.

Overall, it is not clear, what conclusions to draw from the paper. A look at the results shows differences. However, truth is not known, so it is unclear which approach to prefer. For such a conclusion, computer simulations would have to be performed with known parameters. A comparison between results of the two approaches applied to simulated data would reveal which method has more desirable statistical properties (bias, coverage, type I error and power, MSE...)

Major comments:

Regarding the frequentist approach: Please discuss why the SMRW approach was used instead of an IPTW approach.

Regarding the Bayesian approach:

Although from the mathematics nothing would change, the setup for the commensurate prior seems to be reverse compared to the usual setting: one would expect the current control data have the external data mean as prior mean, i.e., γ_1 as $\text{normal}(\gamma_2, 1/\tau)$, not the other way around as stated in the manuscript. Analogously for PFS and OS.

The Bayesian approach heavily depends on the choice of the priors. Hence it is essential that arguments be provided for the selection of priors. In addition, sensitivity analyses have to be performed to investigate the impact of the priors on the results. Specifically the variance of the priors will have a huge impact which has to be assessed. It would be very unusual to rely on just one seemingly arbitrary choice and only state results of this one analysis.

On the same lines: the priors and the posteriors should be shown to assess the amount of prior-data conflict.

Regarding the results shown in Fig 3, it seems as if for DCR the variance increases when external data are borrowed. Please discuss.

Approaches have been proposed to use propensity scores in case of Bayesian borrowing (e.g. Liu M, Bunn V, Hupf B, Lin J, Lin J. Propensity-score-based meta-analytic predictive prior for incorporating real-world and historical data. *Stat Med.* 2021 Sep 30;40(22):4794-4808. doi: 10.1002/sim.9095. Epub 2021 Jun 14. PMID: 34126656).

Dear reviewers,

We very much appreciate your comments, which have helped us to improve our manuscript a lot more. Below you will find our responses to your comments. We hope that you will now find our manuscript more explicit, accurate and complete.

Kind regards,

Chen Li on behalf the authors

Reviewer #1 (Remarks to the Author):

The topic of this paper by Li et al is extremely relevant in the frame of the challenges of clinical research in the era of precision medicine. Indeed the availability of molecular tests able to identify small subgroups of patients with distinct molecular characteristics and hopefully targeted options opens the way to the fragmentation of the disease and the need to re-think about the traditional paradigm of drugs' development. To this regard, I fully embrace the message of this paper, but I also think that by a clinical perspective a few remarks should be made and included in the manuscript.

First of all, as previously mentioned, I do believe that these hybrid-designs might be especially relevant when small subgroups are identified and treated according to the presence of actionable alterations. In this case, having multiple control arms is probably uselessly expensive. However, if we treat patients with the molecular alteration A with the anti-A targeted agent, then we should compare their outcome with the one of patients with the same molecular alteration A treated with a standard option in an historical cohort, especially if this alteration A has a prognostic and not only predictive impact. How should this point be tackled by the proposed approach?

We appreciate the reviewer's comments and acknowledgement for the need of hybrid control design when small subgroups of patients are available. Our study cohort indeed includes MSS metastatic CRC patients who became refractory to the 1st and the 2nd line SOC therapies and to whom no additional BM selection was applied. Patients of the intervention arm are treated with a combination of atezolizumab and isatuximab, which targets the PD-L1 and CD38 proteins, respectively. Atezolizumab is a humanized IgG1 monoclonal antibody, an immune checkpoint inhibitor that binds to receptors of the PD-L1, such as PD-1, and thereby releases "brakes" on the immune system and empowers T cells to kill cancer cells. Isatuximab is also an IgG1-derived monoclonal antibody that binds selectively to a unique epitope on the human surface antigen CD38, and depletion of CD38-

expressing immune cells restores T-cell functions and inhibits immunosuppressive effects of leukocytes. However, in this study, we do not use PD-L1 expression or CD38 as a selection biomarker, i.e. require patients to have specific PD-L1 or CD38 related status. Generally, in the first step to construct an externally derived control cohort, we apply the trial selection in the external data, for example, in this study, we select mCRC patients who receive the third line treatment of regorafenib and present MSS tumours.

While our study does not fit in the scenario that the reviewer describes, where patients with prognostic alteration A are treated with anti-A targeted agents, there are trials that do. For instance, in a recent phase I/II trial of metastatic non-small cell lung cancer (NSCLC), patients who are EGFR Exon 20 insertion-positive are intervened with an EGFR tyrosine kinase inhibitor (mobocertinib), which is designed to selectively target in-frame EGFRex20ins mutations in NSCLC [1]. If a randomised control arm is included in the trial, to implement a hybrid control design, we would first construct the external control cohort by selecting patients with in-frame EGFRex20ins mutations in NSCLC, who also met the trial inclusion and exclusion criteria, and then combine the external control and the trial concurrent control using appropriate statistical models to build the hybrid control arm.

I would also like to see more comments about the choice of factors to be included in the propensity score evaluation. For example in the specific case study presented in the manuscript, among variables included in the propensity score adjustment some of the factors with known impact of patients' prognosis are missing including the number of organs involved, the timing of metastases onset (synchronous vs metachronous), the number of lines of therapy previously received, the presence of peritoneal metastases. How should these variables be chosen and implemented in the propensity score?

We thank the reviewer's comments. Variable selection for propensity score models is indeed an important question to consider when designing the hybrid control, and it becomes trickier when the study sample size is very small, like in our case. In theory, variables that are unrelated to the exposure but related to the outcome, i.e. predictive and/or prognostic variables, should always be included in a propensity score model, whereas variables that are related to the exposure but not or weakly related to the outcome, i.e. instrumental variables, should always be excluded. This is due to a consideration of both bias and variance of the model [2]. In practice, we also need to consider the data availability due to external data involved, and the model convergence due to small sample size. We discussed the possibilities of biases induced due to incomplete inclusion of all potential predictive/prognostic factors that lead to residual confounding and/or unmeasured confounding effects in the third paragraph of the discussion section. Other factors with potential prognostic impact on metastatic colorectal cancer, as the reviewer mentioned in the comment, were not included due to data incompleteness, model nonconvergence, discrepancies of data collection or differences in trial design between the historical trial and the MORPHEUS-CRC trial. In addition, the number of previous lines of therapies was not incorporated given all patients had received 2 lines per protocol inclusion criteria.

Finally, it should be made very clear that efficacy results coming from early phase trials are preliminary by definition and the hybrid design approach may help deriving more clinically relevant information to plan the subsequent steps of drugs' development especially when the feasibility of large-scale trials is hampered by the rarity of some specific conditions rather than a way to "jump" the steps of drugs development towards clinical practice.

We thank the reviewer's comments, and accordingly edited the sentence in the conclusion (p16) to incorporate the reviewer's suggested statement " , although it is not a shortcut to an approval of clinical practice during the process of drug development".

Reviewer #2 (Remarks to the Author):

The authors report results from a randomized trial and use data from one arm of a completed trial to augment the concurrent control arm. As a first step, they select from the historical control arm those patients that would have fulfilled the current eligibility criteria. Then they use a frequentist and a Bayesian method for constructing the so-called hybrid control arm. In the frequentist approach, they use a SMRW propensity approach to incorporate the external data. In the Bayesian approach, the authors use the commensurate prior approach to borrow from the historical data.

The authors address a timely topic. They do not develop new methodology but they apply two existing approaches to a data set and compare the results. In a way, it seems that the authors compare fundamentally different approaches: in the frequentist approach with PS weighing, they take care that the current and the historical arms are similar concerning the patients in regard to the confounders they used for PS calculation. In the Bayesian approach, they borrow irrespective of patient characteristics but adapting to the degree of similarity of the endpoints. Overall, it is not clear, what conclusions to draw from the paper. A look at the results shows differences. However, truth is not known, so it is unclear which approach to prefer. For such a conclusion, computer simulations would have to be performed with known parameters. A comparison between results of the two approaches applied to simulated data would reveal which method has more desirable statistical properties (bias, coverage, type I error and power, MSE...)

We thank the reviewer's comments, and to clarify our conclusions, we performed a simulation study with 1,000 iterations with known parameters and compared the results of the frequentist method and the Bayesian method in terms of their statistical properties, including variance, MSE and 95% coverage. We assumed a historical control sample size of 30 and a trial sample size of 30 with a probability of 0.535 for being assigned to the

experimental arm. Covariates included age, sex, presence of liver metastases, time from metastatic diagnosis to baseline ($>$ vs ≤ 18 months) and ECOG values. Ages in the historical control group and the trial group were assumed to follow normal distributions with mean and standard deviation equivalent to those in the corresponding groups. Other covariates were all binary types and followed binomial distributions with probabilities equivalent to those estimated in the real dataset. We assumed a Weibull baseline hazard function $h_0(t)=\lambda\rho t^{\rho-1}$ with a shape $\rho=1.2$ and a scale $\lambda=0.1$; the cumulative baseline hazard function can then be expressed as $H_0(t)=\lambda t^\rho$. To generate survival time for each patient, we randomly drew v from a uniform distribution $U(0,1)$, and made an inverse transformation of $H_0(t)$: $H_0^{-1}(t)=(t/\lambda)^{1/\rho}$, then survival time t can be computed as $(-\log(v)/\lambda \exp(\mathbf{x}'\beta))^{1/\rho}$, where \mathbf{x}' is a matrix of covariates and β is a vector of coefficients estimated from Cox proportional-hazards model in real data. Censoring time c was assumed to follow an exponential distribution of rate = $\frac{1}{4}$, then the censored survival time was defined as the smaller value of the survival time t and the censoring time c . Results are shown below (Figure 1). In conclusion, the two methods are very similar in terms of model bias and variance, while the coverage probability from the frequentist method is slightly larger than that from the Bayesian method, the Bayesian method shows better precision and accuracy than the frequentist method in this simulation setting, where the sample size is very small, like the early phase trial setting.

Figure 1. Simulation results comparing the frequentist method and the Bayesian method, including coverage, 95% confidence interval width and MSE.

Major comments:

Regarding the frequentist approach: Please discuss why the SMRW approach was used instead of an IPTW approach.

We chose the SMRW as the propensity score weighting method due to a couple of considerations. First, different approaches can result in different estimands of the treatment effects (ICH E9(R1) Addendum). For IPTW, the treatment effect is the population average treatment effect (ATE), hence the results can be generalised to the entire population from which the observed samples are representative of, whereas the SMRW method results in an estimate of the average treatment effect in the treated (ATT), therefore focusing on the patients who would receive the intervention treatment. In randomised controlled trials, ATT and ATE are the same as we assume baseline characteristics and treatment effects in the control and the treated groups are comparable, but in studies that contain non-randomisation components, ATT and ATE estimates can be different, as they target different populations. The choice depends on varied study interests, in medical studies, typically the ATT is used because clinicians often care more about the causal effect of drugs for a targeted group of patients who would potentially receive the drugs.

Second, the propensity score method was originally developed to build external controls in single arm trials, hence there are only two arms (the trial experimental arm and the external control arm) to be weighted. However, in a hybrid control study, we have three arms, the experimental and control arms from the Morpheus trial and the externally derived control arm from the historical IMblaze370 trial. In our propensity score model (i.e., the logistic regression), the externally derived control arm and the experimental arm are used as the treatment indicator (the dependent variable), the internal control arm is not used in the model, but also needs to be weighted. By using the SMRW, both the internal control and experimental arms are given a weight of 1, under the assumption that few confounding effects exist after randomisation, thereby solving the problem of unknown weights for the internal control arm.

Regarding the Bayesian approach:

Although from the mathematics nothing would change, the setup for the commensurate prior seems to be reverse compared to the usual setting: one would expect the current control data have the external data mean as prior mean, i.e., γ_1 as $\text{normal}(\gamma_2, 1/\tau)$, not the other way around as stated in the manuscript. Analogously for PFS and OS.

We thank the reviewer's comments and apologise for the typos. We have now corrected the sentences in the relevant part of the methods, as the reviewer indicated.

The Bayesian approach heavily depends on the choice of the priors. Hence it is essential that arguments be provided for the selection of priors. In addition, sensitivity analyses have to be performed to investigate the impact of the priors on the results. Specifically the variance of the priors will have a huge impact which

has to be assessed. It would be very unusual to rely on just one seemingly arbitrary choice and only state results of this one analysis.

We thank the reviewer’s comments for potential impact of prior choices on the results, especially the variance of the priors. Indeed, to minimise the impact of prior choices on posterior results, we used vague priors that provided little information for the reference amount of borrowing, based on established work for recommendations of priors in previous simulation studies for hybrid control [3]. Moreover, here we performed sensitivity analyses to test if our conclusion holds when variances of the priors change. For each outcome, we tried three prior distribution parameters, including the original ones used in the manuscript (shown in the first column, Table 1). For PFS and OS, the results were consistent when different variances of prior distributions were set; for DCR, the results changed slightly when priors ($1/\tau$) of larger variances were tried, partially due to the very small sample size and thus very large standard deviations of the result estimates.

Table 1. Treatment effect estimates of DCR, PFS and OS for different variances of prior distributions.

	$\tau \sim \text{Gamma}(1,1)$	$\tau \sim \text{Gamma}(1,0.1)$	$\tau \sim \text{Gamma}(1,0.01)$
DCR (OR [95% CI])	0.82 [0.19, 2.91]	1.08 [0.19, 3.56]	1.18 [0.21, 3.86]
	$\tau \sim \text{half-Cauchy}(0, 25)$	$\tau \sim \text{half-Cauchy}(0, 50)$	$\tau \sim \text{half-Cauchy}(0, 100)$
PFS (HR [95% CI])	2.10 [1.03, 3.74]	2.10 [1.05, 3.77]	2.11 [1.04, 3.70]
OS (HR [95% CI])	1.88 [0.82, 3.60]	1.88 [0.83, 3.57]	1.90 [0.85, 3.57]

On the same lines: the priors and the posteriors should be shown to assess the amount of prior-data conflict.

We show here the prior and posterior distributions of beta coefficients for DCR, PFS and OS models. The priors are all standard normal distributions ($\text{norm}^{\sim}(0,1)$), and the posteriors all look normal as well, yet the means are shifted a bit to the right and the variances become smaller.

Figure 2. Prior (blue) and posterior (red) distributions of beta coefficients for DCR, PFS and OS, from left to right.

Regarding the results shown in Fig 3, it seems as if for DCR the variance increases when external data are borrowed. Please discuss.

We thank the reviewer's comment and agree with the observation that the DCR variance was slightly increased when the comparator arm was the hybrid control (β (SD) = -0.20(0.70)) than when the comparator arm was the concurrent control (β (SD) = -0.20(0.65)). However, since the variation is large comparing to the mean of β coefficient, we deemed the DCR variance was similar before and after incorporation of historical trial data, taken the random sampling during iterations into account. Moreover, in the DCR scenario, little was borrowed due to dissimilarity of DCR distributions of the external and the concurrent control arms. We have mentioned the observation in the results "Efficacy assessment" section, where we indicated τ in DCR and in PFS and OS. When evidence for commensurability is weak, τ is forced toward zero, increasing the prior variance of the concurrent-control arm hazards by $1/\tau$, thereby discouraging borrowing from external data. We observed τ was much smaller in DCR than in PFS or OS. We also discussed how similarity of the endpoints from externally derived and concurrent control can influence the amount of borrowing and thus the width of credible sets.

Approaches have been proposed to use propensity scores in case of Bayesian borrowing (e.g. Liu M, Bunn V, Hupf B, Lin J, Lin J. Propensity-score-based meta-analytic predictive prior for incorporating real-world and historical data. Stat Med. 2021 Sep 30;40(22):4794-4808. doi: 10.1002/sim.9095. Epub 2021 Jun 14. PMID: 34126656).

We thank the reviewer's suggestion and add a sentence in the discussion (p13) referring to this paper.

References

1. Zhou C, Ramalingam SS, Kim TM, et al. Treatment Outcomes and Safety of Mobocertinib in Platinum-Pretreated Patients With EGFR Exon 20 Insertion–Positive Metastatic Non–Small Cell Lung Cancer: A Phase 1/2 Open-label Nonrandomized Clinical Trial. *JAMA Oncol*. Published online October 14, 2021. doi:10.1001/jamaoncol.2021.4761
2. Brookhart MA, Schneeweiss S, Rothman KJ, Glynn RJ, Avorn J, Stürmer T. Variable selection for propensity score models. *Am J Epidemiol*. 2006;163(12):1149-1156. doi:10.1093/aje/kwj149
3. Lewis CJ, Sarkar S, Zhu J, Carlin BP. Borrowing from historical control data in cancer drug development: a cautionary tale and practical guidelines. *Stat Biopharm Res*. 2019;11(1):67-78. doi:10.1080/19466315.2018.1497533

Reviewers' comments:

Reviewer #1 (Remarks to the Author):

I am happy with changes made by authors according to my previous review.

Reviewer #2 (Remarks to the Author):

The authors addressed all comments in their rebuttal. Unfortunately, they only made very few changes. As the points this reviewer raised are more or less standard questions that a statistical reader of this paper will have, it is not sufficient to answer only in the rebuttal. Appropriate changes to the manuscript have to be made. The authors should at least briefly address every point also in the main manuscript, potentially pointing to information in the supplemental material.

Dear reviewer 2,

We very much appreciate your additional comments. Below you will find our responses to the comments, as well as the positions where we made updates in the manuscript. We hope that you will now find our manuscript completer and more precise.

Kind regards,

Chen Li on behalf the authors

Reviewer #2 (Remarks to the Author):

The authors addressed all comments in their rebuttal. Unfortunately, they only made very few changes. As the points this reviewer raised are more or less standard questions that a statistical reader of this paper will have, it is not sufficient to answer only in the rebuttal. Appropriate changes to the manuscript have to be made. The authors should at least briefly address every point also in the main manuscript, potentially pointing to information in the supplemental material.

Thank you for the suggestion, we've integrated the materials that we previously only provided in the rebuttal letter into the manuscript / supplements and indicated where and how we made the changes highlighted as below.

Previous comments:

Reviewer #2 (Remarks to the Author):

The authors report results from a randomized trial and use data from one arm of a completed trial to augment the concurrent control arm. As a first step, they select from the historical control arm those patients that would have fulfilled the current eligibility criteria. Then they use a frequentist and a Bayesian method for constructing the so-called hybrid control arm. In the frequentist approach, they use a SMRW propensity approach to incorporate the external data. In the Bayesian approach, the authors use the commensurate prior approach to borrow from the historical data.

The authors address a timely topic. They do not develop new methodology but they apply two existing approaches to a data set and compare the results. In a way, it seems that the authors compare fundamentally different approaches: in the frequentist approach with PS weighing, they take care that the current and the historical arms are similar concerning the patients in regard to the confounders

they used for PS calculation. In the Bayesian approach, they borrow irrespective of patient characteristics but adapting to the degree of similarity of the endpoints. Overall, it is not clear, what conclusions to draw from the paper. A look at the results shows differences. However, truth is not known, so it is unclear which approach to prefer. For such a conclusion, computer simulations would have to be performed with known parameters. A comparison between results of the two approaches applied to simulated data would reveal which method has more desirable statistical properties (bias, coverage, type I error and power, MSE...)

We thank the reviewer's comments, and to clarify our conclusions, we performed a simulation study with 1,000 iterations with known parameters and compared the results of the frequentist method and the Bayesian method in terms of their statistical properties, including variance, MSE and 95% coverage. We assumed a historical control sample size of 30 and a trial sample size of 30 with a probability of 0.535 for being assigned to the experimental arm. Covariates included age, sex, presence of liver metastases, time from metastatic diagnosis to baseline ($>$ vs ≤ 18 months) and ECOG values. Ages in the historical control group and the trial group were assumed to follow normal distributions with mean and standard deviation equivalent to those in the corresponding groups. Other covariates were all binary types and followed binomial distributions with probabilities equivalent to those estimated in the real dataset. We assumed a Weibull baseline hazard function $h_0(t) = \lambda \rho t^{\rho-1}$ with a shape $\rho = 1.2$ and a scale $\lambda = 0.1$; the cumulative baseline hazard function can then be expressed as $H_0(t) = \lambda t^\rho$. To generate survival time for each patient, we randomly drew v from a uniform distribution $U(0,1)$, and made an inverse transformation of $H_0(t)$: $H_0^{-1}(t) = (t/\lambda)^{1/\rho}$, then survival time t can be computed as $(-\log(v)/\lambda \exp(\mathbf{x}'\beta))^{1/\rho}$, where \mathbf{x}' is a matrix of covariates and β is a vector of coefficients estimated from Cox proportional-hazards model in real data. Censoring time c was assumed to follow an exponential distribution of rate = $1/4$, then the censored survival time was defined as the smaller value of the survival time t and the censoring time c . Results are shown below (Figure 1). In conclusion, the two methods are very similar in terms of model bias and variance, while the coverage probability from the frequentist method is slightly larger than that from the Bayesian method, the Bayesian method shows better precision and accuracy than the frequentist method in this simulation setting, where the sample size is very small, like the early phase trial setting. The simulation work was added in the supplement and mentioned in the discussion on page 14.

Figure 1. Simulation results comparing the frequentist method and the Bayesian method, including coverage, 95% confidence interval width and MSE.

Major comments:

Regarding the frequentist approach: Please discuss why the SMRW approach was used instead of an IPTW approach.

We chose the SMRW as the propensity score weighting method due to a couple of considerations. First, different approaches can result in different estimands of the treatment effects (ICH E9(R1) Addendum). For IPTW, the treatment effect is the population average treatment effect (ATE), hence the results can be generalised to the entire population from which the observed samples are representative of, whereas the SMRW method results in an estimate of the average treatment effect in the treated (ATT), therefore focusing on the patients who would receive the intervention treatment. In randomised controlled trials, ATT and ATE are the same as we assume baseline characteristics and treatment effects in the control and the treated groups are comparable, but in studies that contain non-randomisation components, ATT and ATE estimates can be different, as they target different populations. The choice depends on varied study interests, in medical studies, typically the ATT is used because clinicians often care more about the causal effect of drugs for a targeted group of patients who would potentially receive the drugs.

Second, the propensity score method was originally developed to build external controls in single arm trials, hence there are only two arms (the trial experimental arm and the external control arm) to be weighted. However, in a hybrid control study, we have three

arms, the experimental and control arms from the Morpheus trial and the externally derived control arm from the historical IMblaze370 trial. In our propensity score model (i.e., the logistic regression), the externally derived control arm and the experimental arm are used as the treatment indicator (the dependent variable), the internal control arm is not used in the model, but also needs to be weighted. By using the SMRW, both the internal control and experimental arms are given a weight of 1, under the assumption that few confounding effects exist after randomisation, thereby solving the problem of unknown weights for the internal control arm. The rationale for choosing the SMRW over the IPTW work was added in the supplement and mentioned in the method on page 10.

Regarding the Bayesian approach:

Although from the mathematics nothing would change, the setup for the commensurate prior seems to be reverse compared to the usual setting: one would expect the current control data have the external data mean as prior mean, i.e., γ_1 as $\text{normal}(\gamma_2, 1/\tau)$, not the other way around as stated in the manuscript. Analogously for PFS and OS.

We thank the reviewer's comments and apologise for the typos. We have now corrected the sentences in the relevant part of the methods, as the reviewer indicated.

The Bayesian approach heavily depends on the choice of the priors. Hence it is essential that arguments be provided for the selection of priors. In addition, sensitivity analyses have to be performed to investigate the impact of the priors on the results. Specifically the variance of the priors will have a huge impact which has to be assessed. It would be very unusual to rely on just one seemingly arbitrary choice and only state results of this one analysis.

We thank the reviewer's comments for potential impact of prior choices on the results, especially the variance of the priors. Indeed, to minimise the impact of prior choices on posterior results, we used vague priors that provided little information for the reference amount of borrowing, based on established work for recommendations of priors in previous simulation studies for hybrid control [3]. Moreover, here we performed sensitivity analyses to test if our conclusion holds when variances of the priors change. For each outcome, we tried three prior distribution parameters, including the original ones used in the manuscript (shown in the first column, Table 1). For PFS and OS, the results were consistent when different variances of prior distributions were set; for DCR, the results changed slightly when priors ($1/\tau$) of larger variances were tried, partially due to the very small sample size and thus very large standard deviations of the result estimates. This sensitivity analysis was added in the supplement and mentioned in the method on page 10.

On the same lines: the priors and the posteriors should be shown to assess the amount of prior-data conflict.

We show here the prior and posterior distributions of beta coefficients for DCR, PFS and OS models. The priors are all standard normal distributions ($\text{norm}\sim(0,1)$), and the posteriors all look normal as well, yet the means are shifted a bit to the right and the variances become smaller. **This result was added in the supplement and mentioned in the method on page 10.**

Figure 2. Prior (blue) and posterior (red) distributions of beta coefficients for DCR, PFS and OS, from left to right.

was slightly increased when the comparator arm was the hybrid control ($\beta(\text{SD}) = -0.20(0.70)$) than when the comparator arm was the concurrent control ($\beta(\text{SD}) = -0.20(0.65)$). However, since the variation is large comparing to the mean of β coefficient, we deemed the DCR variance was similar before and after incorporation of historical trial data, taken the random sampling during iterations into account. Moreover, in the DCR scenario, little was borrowed due to dissimilarity of DCR distributions of the external and the concurrent control arms. We have mentioned the observation in the results “Efficacy assessment” section, where we indicated τ in DCR and in PFS and OS. When evidence for commensurability is weak, τ is forced toward zero, increasing the prior variance of the concurrent-control arm hazards by $1/\tau$, thereby discouraging borrowing from external data. We observed τ was much smaller in DCR than in PFS or OS. We also discussed how similarity of the endpoints from externally derived and concurrent control can influence the amount of borrowing and thus the width of credible sets.

Approaches have been proposed to use propensity scores in case of Bayesian borrowing (e.g. Liu M, Bunn V, Hupf B, Lin J, Lin J. Propensity-score-based meta-analytic predictive prior for incorporating real-world and historical data. Stat Med. 2021 Sep 30;40(22):4794-4808. doi: 10.1002/sim.9095. Epub 2021 Jun 14. PMID: 34126656).

We thank the reviewer’s suggestion and add a sentence in the discussion (p13) referring to this paper.

REVIEWERS' COMMENTS:

Reviewer #2 (Remarks to the Author):

Thank you for including the material in the manuscript and/or supplement.